# Estimating Soil Salinity with Different Levels of Vegetation Cover by Using Hyperspectral and Non-Negative Matrix Factorization Algorithm

**DOI:** 10.3390/ijerph20042853

**Published:** 2023-02-06

**Authors:** Jianfei Cao, Han Yang, Jianshu Lv, Quanyuan Wu, Baolei Zhang

**Affiliations:** 1College of Geography and Environment, Shandong Normal University, Jinan 250014, China; 2Zhongke Shandong Dongying Institute of Geographic Sciences, Dongying 257000, China

**Keywords:** partially vegetated surfaces, mixed hyperspectra, non-negative matrix factorization, soil salt content

## Abstract

Hyperspectral technology has proven to be an effective method for monitoring soil salt content (SSC). However, hyperspectral estimation capabilities are limited when the soil surface is partially vegetated. This work aimed to (1) quantify the influences of different fraction vegetation coverage (FVC) on SSC estimation by hyperspectra and (2) explore the potential for a non-negative matrix factorization algorithm (NMF) to reduce the influence of various FVCs. Nine levels of mixed hyperspectra were measured from simulated mixed scenes, which were performed by strictly controlling SSC and FVC in the laboratory. NMF was implemented to extract soil spectral signals from mixed hyperspectra. The NMF-extracted soil spectra were used to estimate SSC using partial least squares regression. Results indicate that SSC could be estimated based on the original mixed spectra within a 25.76% FVC (R^2^_cv_ = 0.68, RMSE_cv_ = 5.18 g·kg^−1^, RPD = 1.43). Compared with the mixed spectra, NMF extraction of soil spectrum improved the estimation accuracy. The NMF-extracted soil spectra from FVC below 63.55% of the mixed spectra provided acceptable estimation accuracies for SSC with the lowest results of determination of the estimation R^2^_cv_ = 0.69, RMSE_cv_ = 4.15 g·kg^−1^, and RPD = 1.8. Additionally, we proposed a strategy for the model performance investigation that combines spearman correlation analysis and model variable importance projection analysis. The NMF-extracted soil spectra retained the sensitive wavelengths that were significantly correlated with SSC and participated in the operation as important variables of the model.

## 1. Introduction

As one of the main forms of soil degradation, soil salinization seriously restricts the efficient utilization of soil resources and agricultural development [1,2]. For example, the Yellow River Delta exhibits obvious ecological vulnerability and faces a severe soil salinization problem due to frequent land–sea interactions [3,4]. Dynamic monitoring of soil salinization is key to the sustainable development of the ecological environment and the effective management of soil resources [5,6].

Recently, the potential relationship between spectral values and soil salinity has been the basic theory for estimating soil salinity by the hyperspectral technique [7,8,9]. However, mixed hyperspectral limits the accuracy of soil salt content (SSC) assessments in areas where the soil surface is partially covered with vegetation [10]. To deal with the interference of vegetation, some vegetation indices, such as the ratio vegetation index (RVI), the normalized difference vegetation index (NDVI), and the soil-adjusted vegetation index (SAVI), were used to evaluate SSC by constructing a relational model [11,12]. However, vegetation indices cannot always perform well due to their indirect detection of SSC [13,14].

In addition, spectral unmixing, as an effective spectrum, has recently been used to solve the problem of mixed pixels in the remote sensing monitoring of soil property. Bartholomeus et al. (2011) [15] used residual spectral unmixing (RSU) to alleviate the interference of maize and obtained the bare soil spectrum from a mixed spectrum to estimate soil organic matter. However, RSU methods require prior knowledge of soil and vegetation proportions as well as spectral information, which is usually difficult to obtain, resulting in a limited application. Blind source separation (BSS) technology made up for the insufficient application of the RSU method and did not require any prior knowledge of the source spectrum and its mixing mode. Independent component analysis (ICA) was used to extract soil spectra from mixed spectra to further estimate clay content in some vegetated areas [16]. As another BBS, NMF is not limited to independent source signals and has a non-negative constraint on extracted source spectra [17]. Some scholars have confirmed that NMF could be successfully used to extract soil attribute characteristics from mixed spectra, and its performance was better than ICA and RSU [18,19,20]. However, we found that NMF capacity was often assumed to be consistent in the above applications. In most salinized areas, especially coastal areas, vegetation coverage is not uniform due to the different salinity tolerances and phenological laws of different vegetation [21]. How do different vegetation coverage levels affect the spectral response and estimation accuracy of SSC? What is the impact on the application performance of NMF? Few studies have adequately answered these questions.

At present, some studies have attempted to explain the mechanism of spectral prediction models to understand the process of spectral prediction more clearly. Correlation analysis was used to understand the significant correlation between two variables and was a common spectral analysis method in the quantitative prediction of remote sensing. In addition, as the most commonly used spectral model, variable importance in the projection (VIP) analysis can identify the important variables in the PLSR’s model calibrations. The sensitive wavelength of correlation analysis and important wavelength obtained by VIP were considered able to explain the mechanism of the spectral prediction [22]. Jiang et al. (2016) [23] explained the influence mechanism of soil moisture in the spectral prediction of soil organic matter. Chen et al. (2019) [24] explored the indirect effects of Fe compounds and organic matter in the prediction mechanism of soil heavy metals. However, this method has rarely been used in research into the estimation of SSC with mixed disturbance of partial vegetation.

In this paper, we attempted to explore the potential of the NMF method for hyperspectral SSC estimation at different FVCs and explain its performance. A laboratory simulated experiment was conducted to obtain nine FVC levels of mixed spectra. Partial least squares regression (PLSR) models were used to investigate the accuracy of the estimation of SSC at different FVC levels. Then, NMF was used to extract soil endmembers from the mixed spectrum for comparative analysis. The main objectives of this study were to (1) explore the impact of mixed spectra on soil salinity estimation with different FVCs, (2) evaluate the performance of NMF-extracted soil spectra in SSC estimation, and (3) explain the performance by combining correlation analysis and model variable importance projection analysis.

## 2. Materials and Methods

### 2.1. Data Sources

#### 2.1.1. Study Site and Soil Sampling

The study area was set in the Yellow River Delta (37°05′–37°35′ N, 118°20′–118°50′ E) in the northeast of Shandong Province, China (Figure 1). Yellow River Delta is the largest coastal wetland in North China and a typical coastal salinization area. The Yellow River Delta has an average annual temperature of 12.1 °C and an average annual rainfall of 551.6 mm. It is a typical warm temperate monsoon climate. At present, soil salinization has become an important environmental problem restricting local ecological stability and agricultural development [25]. Most of the natural vegetation is salt-tolerant, mainly reed, Pterocarpus, Swertia, white grass, Tamarix, etc. [26]. The soil type of the Yellow River Delta was affected by modern Yellow River flooding deposition and formed coastal tidal soil [27].

A total of 208 surface soils were sampled from June to July 2020, and the land use types included cultivated land, unused land, and grassland. For each sample, five subsamples were collected using the double diagonal method. After mixing each fraction, 1 kg of soil sample was put into the sample bag by the four-part method, and the excess part was discarded. All samples were 0–20 cm deep topsoil samples. The geographic locations of each sample were recorded using a GPS receiver. All the sieved samples were thoroughly mixed and divided into two parts using the quartering method: one for soil spectral measurement and the other for soil property determination.

#### 2.1.2. Laboratory Analysis

The soil collected back to the laboratory was dried by air drying method. The dried soil sample was laid flat on the board, and the ground soil was then sifted through a screen with 2 mm sieve holes [28]. The soil water-soluble salts were extracted by equilibrium leaching method according to the ratio of soil to water 1:5, and then the total salt content and the content of eight main ions such as CO_3_^2−^, HCO_3_^−^, Cl^−^, SO_4_^2−^, Ca^2+^, Mg^2+^, Na^+^, K^+^ in the leaching solution were determined [29,30]. Soil pH was measured using an electrometric method on a soil/water suspension. Soil organic matter was measured using the Walkley–Black method [27]. To improve the precision of the results, every sample was analyzed three times, and an average value was calculated.

### 2.2. Experiments

A laboratory simulated experiment was conducted to investigate the impact of the mixed spectra on SSC estimation. The experiment was a factorial design with two independent variables (SSC and FVC) and one dependent variable (spectral data). Fresh wheat leaves in the Yellow River Delta were collected and transported to the laboratory, where they simulated the typical vegetation cover for mixed scenes. In the laboratory, the 208 samples were dried to a constant weight and ground to a fine powder with a particle size of less than 0.149 mm. Among them, 140 samples were used as calibration sets, and 68 samples were used as validation sets.

(1)Soil sample preparation: As shown in Figure 2, the soil was placed into the plastic tray (7.5 cm radius). It was filled with approximately 150 g of soil at a depth of approximately 3 cm.(2)Vegetation cover simulation: Wheat leaves were cut into 3 cm sections (1 cm height) and placed on the soil surface in the scene. Without damaging the soil sample, 3 prepared leaves were added at a time. Digital photographs of each scene were taken using a digital camera for the vegetation coverage calculation [28]. The FVC was calculated using the method proposed by Zhang et al. (2013) [29]. For each soil sample, nine levels of mean FVC were designed as follows: 5.61%, 9.21%, 16.34%, 25.76%, 37.81%, 48.28%, 56.42%, 63.55%, and 76.42%.(3)Spectrum measurement: The spectrometer was placed vertically approximately 15 cm above the soil sample with a 25° field of view. The reflectance of the mixed scenes was measured within a radius of approximately 3.3 cm (Figure 2). Spectra were measured with an ASD FieldSpec HandHeld 3 portable spectroradiometer (Analytical Spectra Devices, Inc., Boulder, CO, USA), which covered the wavelengths of 350–2500 nm with a sampling interval of 1.4 nm (350–1000 nm) and 2 nm (1000–2500 nm) [30]. Soil samples were illuminated using two 50 W halogen lamps with an incident angle of 45°, which were positioned 50 cm from the sample [31,32]. The spectrometer probe was positioned vertically, approximately 30 cm above the sample. The spectrometer was preheated for 20–30 min and calibrated with a white panel before each measurement [27]. Ten spectral measurements were repeated for each soil sample [31]. As a result, a total of 9 × 208 × 10 = 18,720 mixed spectra were obtained. Spectral preprocessing was performed using ASD software to remove physical variability from light scattering and highlight the spectral features of interest [33]. The spectra were subjected to Savitzky–Golay smoothing with a moving window width of nine and transformed into the first derivative. Because of the presence of high-frequency noise at the edges of the spectrum, after the preprocessing stage, all the spectra were reduced to 500–2350 nm.

### 2.3. Methodology

A flowchart that shows soil salinity estimation with different levels of vegetation cover is shown in Figure 3.

#### 2.3.1. Non-Negative Matrix Factorization

NMF was one of the separation algorithms of BSS, which was applied to extract soil spectra from different FVC-simulated mixed scenes in this study [34].

Given an original matrix ***X*** ∈ ***R****^n^*^×*m*^, the goal of NMF was to find two non-negative matrices, A ∈ ***R****^n^*^×*r*^ and S ∈ ***R****^r^*^×*m*^ [35]. The equation can be formulated as follows:(1)$$X\approx AS,\;a_{j}\approx AS_{j}$$
where matrix A is the endmember matrix, each column of which is called the potential basic element. Matrix S is the abundance matrix, each column of which corresponds to the matrix X. *r* is the constant of the decomposition rank in the non-negative matrices, generally determined from (*n* + *m*) *r* < *nm*. Specifically, m is the number of pixels, and *n* is the number of hyperspectral bands. In the current study, the mixed spectra were mainly soil and vegetation; thus, the number of source spectra r was set as 2.

The following cost function can be used to solve the NMF problem:(2)fA,S min(A,S)=12||X−AS||F2s.t.A≥0, S≥0, 1ST=1mT

The loss function expressed in (2) should minimize the gap between A and S and the original matrix ***X***. The abundance sum to one constraint (ASC) should be considered simultaneously, that is, $1_{S}^{T}=1_{m}^{T}$ [36]. In this method, the new observation and signature matrices are defined as follows:(3)$$\hat{X}\left[\begin{array}{c} X\\ \delta 1_{m}^{T} \end{array}\right],\;\hat{A}=\left[\begin{array}{c} A\\ \delta 1_{r}^{T} \end{array}\right]$$
where δ > 0 was used to adjust abundance to accomplish ASC constraints. The NMF method was implemented by MATLAB R2015 (The MathWorks, Inc., Natick, MA, USA).

#### 2.3.2. Model Calibration and Validation

Partial least squares regression (PLSR) is a multivariate statistical method and has been proven to be an effective linear regression technique for estimating soil properties using spectral data [37]. In this study, PLSR was applied to model the NMF-extracted soil spectra (independent variable) and SSCs (dependent variable). The leave-one-out cross-validation method was applied to the PLSR process [33]. The PLSR method was implemented by Unscrambler X 10.4 (Computer Aided Modelling, Trondheim, Norway).

To investigate the influence of FVC on the estimation of SSC, two model validation strategies were adopted, named levels’ validation and multiple validations, respectively. The level validation was used to study NMF performance in evaluating SSC with the same FVC level. Multiple validations were used to study NMF performance in evaluating SSC with datasets that were randomly chosen from nine FVC levels.

The ratio of performance to deviation (RPD), the determination coefficient (R^2^), and root mean square error (RMSE) were used to evaluate the accuracy of the soil salinity model [38,39]. According to studies by Chang et al. (2001) and Terhoeven-Urselmans et al. (2010) [40,41], RPD values were interpreted in three levels: RPD > 2.0 indicates a good estimate, 1.4 < RPD < 2.0 represents a satisfied estimate, and RPD < 1.4 indicates a poor estimation.

#### 2.3.3. Estimation Mechanism Analysis

To explore the potential of the NMF method for SSC estimation and explain the estimation mechanism, a strategy of combining spearman correlation analysis (SCA) and model variable importance projection (VIP) analysis was proposed. Spearman correlation coefficients were calculated to indicate the relationships among SSC, original mixed spectra, and NMF-extracted soil spectra [21]. On the other hand, the important wavelengths in PLSR modeling were identified by the VIP value. By comparing the correlation coefficient and the important wavelength in PLSR, the influence of mixed pixels on estimation and NMF performance was analyzed. VIP values greater than 1 were considered important variables, corresponding to the selection of important spectral wavelengths [42,43].

## 3. Results and Discussion

### 3.1. Analysis of Spectral Characteristic Analysis

The average reflectance of the bare soil with different SSCs is shown in Figure 4. The soil spectra of different SSCs had similar overall morphology but different spectral intensities. In the range of 500–1410 nm, the reflectance tended to increase rapidly with increasing wavelength. When the wavelengths were longer than 1410 nm, the spectral curves fluctuated, and the positions of the peaks and troughs were relatively stable. The shoulder features at around 800 nm were caused by hydroxyl ions and water within the lattice of the hydrated minerals [44,45]. The diagnostic absorption features near 1410, 1940, and 2210 nm were considered the combined result of O–H stretching and H–O–H bending fundamental sand overtone [46].

To explore the influences of FVC and SSC on the soil spectra, the average spectra of soil samples were obtained under two single-variable conditions. Figure 5 shows the changes in salty soil spectra with different FVCs. There was a decline in reflectance across the entire wavelength domain with increasing FVC for the same SSC. The above variation was more apparent for longer wavelengths, especially in the spectral absorption bands (1410, 1940, and 2210 nm). The spectral green, red, and red edge features were more pronounced in the spectra with higher FVCs. From the observation point of view, the magnitude of the influence of vegetation on the soil spectrum was much higher than that of salinity.

### 3.2. Estimation SSC with Different Levels of Vegetation

Table 1 shows the PLSR model accuracy constructed by simulating mixed spectra and bare soil spectra. The calibration and validation results of the SSC estimation decreased as the FVC increased. The best accuracy of the estimation results was obtained for the bare soil spectra (R^2^_cv_ = 0.95, RMSE_cv_ = 2.55 g·kg^−1^) followed by 5.61% FVC-simulated mixed sample (R^2^_cv_ = 0.82, RMSE_cv_ = 3.72 g·kg^−1^) and 9.21% FVC-simulated mixed sample (R^2^_cv_ = 0.77, RMSE_cv_ = 4.06 g·kg^−1^). In terms of RPD, good estimations were obtained with an FVC of 5.61% (RPD = 2.02). Satisfactory estimations were obtained with FVC values of 9.21%, 16.34%, and 25.76% (RPD = 1.84, 1.53, and 1.43, respectively). Overall, these results indicate that SSC could be estimated based on the original mixed spectra within a 25.76% FVC.

### 3.3. Performance of NMF

#### 3.3.1. Performance of Soil Spectra Extraction

NMF has been used previously in the estimation of soil salinity at partially vegetation cover areas by remote sensing. Ouerghemmi et al. (2016) [16] demonstrated the NMF approach was effective for moderate vegetation coverage (NDVI < 0.55) using VNIR/SWIR hyperspectral airborne data. Liu et al. (2019) successfully applied NMF to separate soil spectral signals from mixed pixels of Landsat 5 TM to estimate the soil salinity in a partially vegetated area (FVC ≤ 58.26%). However, the effectiveness of NMF for hyperspectral SSC estimation at different FVCs was not quantitatively analyzed. In the current research, the saline soil samples contained vegetation coverage ranging from 0% to 76.42% (nine groups of mean FVC were designed as follows: 5.61%, 9.21%, 16.34%, 25.76%, 37.81%, 48.28%, 56.42%, 63.55%, and 76.42%).

The NMF-extracted spectra from mixed spectra are shown in Figure 6. Results showed that two distinguishing spectra were extracted from each original mixed spectrum. The two spectra extracted by NMF are similar to the soil and vegetation spectra, respectively. This result indicated that the NMF decomposition process did not suffer from large deviations. When the vegetation coverage was 5.61%, the shape of the extracted soil spectral was consistent with that of the bare soil spectral. When the vegetation coverage was greater than 9.21%, the extracted soil spectra gradually retained the vegetation red edge feature at about 700 nm. With the increase in FVC, the representative vegetation characteristics were much more noticeable, such as the reflection peak at 550 nm and the absorption valley at 750 nm, 1410, 1940, and 2210 nm. When the FVC was higher than 63.55%, the shape of the extracted soil spectra was very similar to that of the mixed spectra. The results indicated that the performance of NMF to extract soil spectra worsened with increasing FVC, mainly because weak soil information could not be effectively captured.

#### 3.3.2. Performance of SSC Estimation

The PLSR model of SSC was established on NMF-extracted soil spectra with nine levels of FVC, and the results are shown in Table 2. The estimation accuracy was improved compared with the accuracy based on the original mixed spectra (Table 2). For NMF-extracted soil spectra with FVC in the range of 5.61–48.28%, the accuracy of estimating SSC was almost similar, with R^2^_cv_ = 0.79–0.88 and RMSE_cv_ = 3.27–3.89 g·kg^−1^. When the FVC was higher than 56.42%, the accuracy of estimation was reduced rapidly, with R^2^_cv_ = 0.74 and RMSE_cv_ = 3.65 g·kg^−1^. The soil spectra extracted from 76.42% FVC had the worst estimation, with R^2^_cv_ = 0.36 and RMSE_cv_ = 5.54 g·kg^−1^. In terms of RPD, good estimations were obtained with FVC values of 25.76%, 48.28%, and 56.42% (RPD = 2.30, 2.03, and 2.05, respectively). Satisfactory estimations were obtained with FVC values of 5.61%, 9.21%, 16.34%, 37.81%, and 63.55% (RPD = 1.93, 1.98, 1.94, 1.89, and 1.80, respectively). These results emphasize that soil spectra extracted from the FVC below 63.55% using NMF have a substantial ability to estimate SSC. When the FVC increased to 76.42%, the NMF-extracted soil spectra could not be used for the high-accuracy estimation of SSC (RPD = 1.37 < 1.4).

Calibration and validation samples were randomly chosen from nine levels of mixed experiments and used to verify the performance of NMF. The results of estimation accuracy with multiple FVCs were calculated, as shown in Table 3. In terms of the mixed spectra, the accuracy of estimating SSC was not efficient, with R^2^_cv_ = 0.69, RMSE_cv_ = 7.24 g·kg^−1^, and RPD = 1.68. For the soil spectrum extracted by NMF, the estimation accuracy of SSC was significantly improved, with R^2^_cv_ = 0.84, RMSE_cv_ = 5.99 g·kg^−1^, and RPD = 2.04. Compared with the mixed spectra, NMF extraction of soil spectrum improved the estimation accuracy.

According to the validation results, NMF can effectively weaken the influence of vegetation cover on salinity estimation in the laboratory mixed simulation spectrum. The validation quantitatively analyzed the performance of NMF under different vegetation coverage levels. With an increase in FVC, the ability of NMF to remove mixing inference weakens gradually, and the estimation accuracy decreases more and more rapidly. The soil spectra extracted by NMF from FVC below 63.55% had a significant ability to estimate SSC, with R^2^_cv_ ≥ 0.69, RMSE_cv_ ≤ 4.15 g·kg^−1^, and RPD ≥ 1.8.

In this study, mixed spectra were obtained from simulation collected experiments. As an unsupervised method, NMF extracted the soil endmember spectra through repeated iterations and did not require prior knowledge. However, the parameters of NMF have a restriction that the number of source spectra < min (the number of original mixed spectra, the number of bands) [47]. It was challenging to obtain multiple related original spectra for remote sensing data in practical applications. Liu et al. (2019) [19] proposed to utilize the spectral angle method to select multiple original mixture spectra from the surrounding four pixels. However, this is limited by the spatial resolution of the pixel and the surrounding vegetation cover variability of the sampling sites. Therefore, it seems to be very important to investigate vegetation coverage in future large-scale airborne or space-borne spectroscopic applications. In addition, the actual remote sensing application was also affected by other factors (soil moisture, other mixed features, image spatial resolution, etc.), which makes the values in this study not applicable. To further generalize this study, we will conduct field in situ experiments for exploration in the future.

### 3.4. Estimation Mechanism Analysis

#### 3.4.1. Explanation of Influence Mechanism

In this study, a strategy of combining SCA and model VIP was proposed to explain the estimation mechanism. SCA revealed a change in significant wavelengths of soil salinity with a change in vegetation coverage in soil samples. On this basis, we compared the VIP important wavelength of the PLSR model with the significant wavelengths of SCA. The results are shown in Figure 7. Spearman correlation coefficients (r) were used to reflect the correlation between spectral reflectance and SSC. The absolute correlation coefficient (|r|) was higher than 0.4 (*p* < 0.01), which is generally considered closely related and considered a significant wavelength [48,49]. With a higher |r|, the corresponding wavelength is more sensitive to the soil salt. The VIP value indicates the relative importance of wavelengths in the SSC estimation model, and the important wavelengths are marked in gray. In terms of Spearman correlations, the significant wavelengths of soil salt were approximately 800, 1410, 1940, and 2210 nm, which were similar to the findings of Quan et al. (2012) [50]. This was primarily due to the use of coastal tidal soil samples, which mainly contain the soluble salts NaCl and MgCl_2_ (Weng et al., 2010). With an increase in FVC, the correlation coefficient curve tends to decrease as a whole, and significant wavelengths gradually narrow or disappear. This means that SSC information becomes difficult to extract in the presence of vegetation cover. According to the VIP, the important wavelengths were similar to the significant wavelengths corresponding to the correlation coefficients. However, the number of wavelengths that fell within the gray area decreased and dispersed with increasing FVC. This can be attributed to the important wavelengths that would not be accurately identified for modeling because of the interference of vegetation cover. Moreover, the mixed disturbance of vegetation cover increases the redundant variables involved in the calculation and reduces the accuracy of the SSC estimation model.

#### 3.4.2. Explanation of Optimization Mechanism

The soil hyperspectra extracted by NMF were analyzed using SCA and VIP, and the results are shown in Figure 8. Compared with mixed spectra (Figure 7), the Spearman correlations of certain wavelengths (e.g., near 800, 1410, 1940, and 2210 nm) were significantly increased, and the shape of the correlation coefficient curve was more similar to that of bare soil after NMF spectral separation. However, this change was not significant when the FVC was higher than 63.55%. From the perspective of VIP, a series of important wavelengths were identified again, and more important wavelengths coincided with the sensitive wavelengths of SCA based on NMF-extracted soil spectra. The results showed that the NMF-extracted soil spectra retained the effective spectral characteristics of the soil salt. This proved the superior potential of NMF in alleviating the influence of vegetation cover in SSC estimation. However, the decomposition ability gradually decreased with increasing FVC. This is mainly because the solution space of the NMF method is not unique, and the soil salt-sensitive information is covered in the solution process [51,52,53,54].

All in all, the correlation between spectrum and soil salinity was weakening, and the important wavelengths of the PLSR model became dispersed with increasing FVC. For the soil salinity estimation model, the significant wavelengths would not be accurately identified and the redundant variables involved in the calculation. After the mixed pixels were decomposed, the NMF-extracted soil spectra retained the sensitive wavelengths that were significantly correlated with SSC and participated in the operation as important variables of the model. Our analysis of the estimation mechanism suggested that selecting remote sensing images containing sensitive wavelengths of soil salt will be more effective.

## 4. Conclusions

This study applied the hyperspectral and non-negative matrix factorization (NMF) algorithm to estimate the soil salinity of coastal soil at different levels of vegetation cover. Additionally, we proposed a strategy for the model performance investigation that combined SCA and VIP analysis. The main results can be summarized as follows:

The reflectance decreased with increasing FVC under the same SSC. The most specific characteristics of soil salt were changed due to the influences of vegetation cover on the soil surface hyperspectra.

From the perspective of soil spectra extraction, NMF was highly effective until the FVC reached 25.76%. When the FVC exceeded 30%, vegetation response characteristics gradually occurred in the extracted soil spectra and became more pronounced with increasing FVC. From the perspective of SSC estimation, acceptable accuracies were obtained with an FVC below 63.55% derived from the NMF-extracted soil spectra, with the lowest results of R^2^_cv_ = 0.69, RMSE_cv_ = 4.15 g·kg^−1^, and RPD = 1.8.

The mixed spectrum of vegetation cover decreased the correlation between reflectance spectra and SSC in the entire wavelength domain. In contrast, the NMF-extracted soil spectra retained the sensitive wavelengths, which were significantly correlated with SSC and participated in the operation as important variables of the model. This study demonstrated the potential of NMF in reducing the influence of different FVCs based on laboratory mixed spectra.

## Figures and Tables

**Figure 1 ijerph-20-02853-f001:**
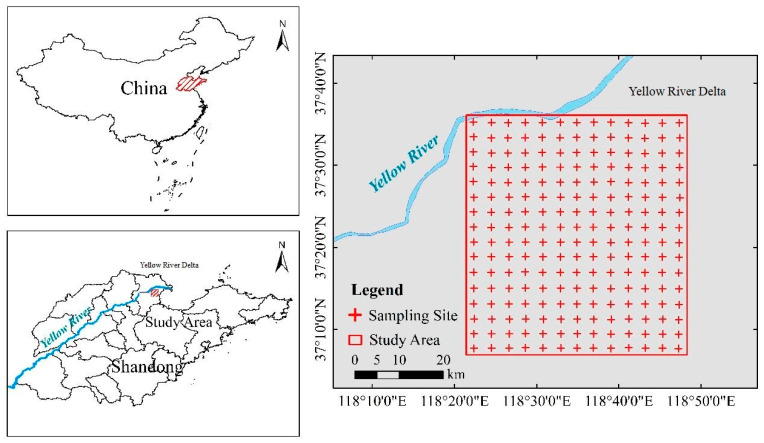
Location of the study area and spatial distribution of sampling sites.

**Figure 2 ijerph-20-02853-f002:**
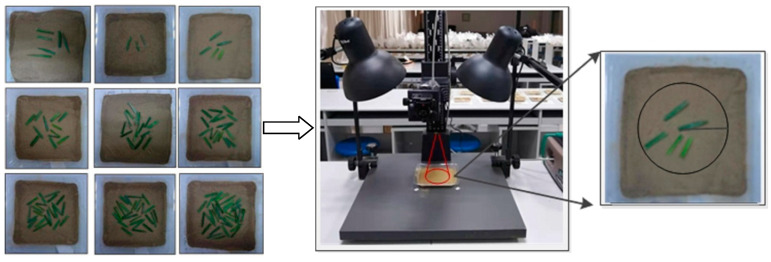
Mixed scenes simulation and spectral measurement.

**Figure 3 ijerph-20-02853-f003:**
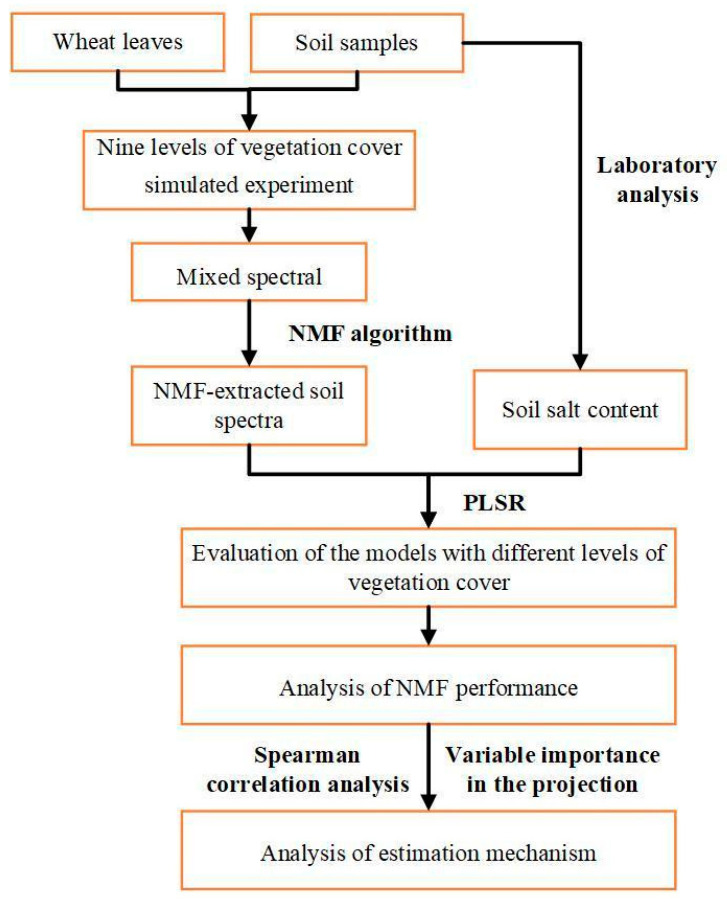
Flowchart of the soil salinity estimation with different levels of vegetation cover by NMF.

**Figure 4 ijerph-20-02853-f004:**
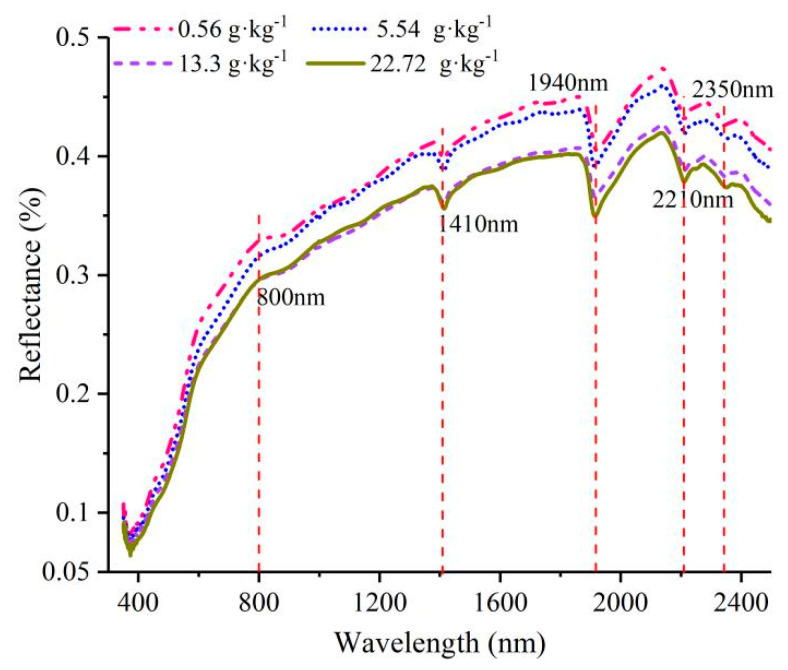
Average reflectance of soil with different SSCs.

**Figure 5 ijerph-20-02853-f005:**
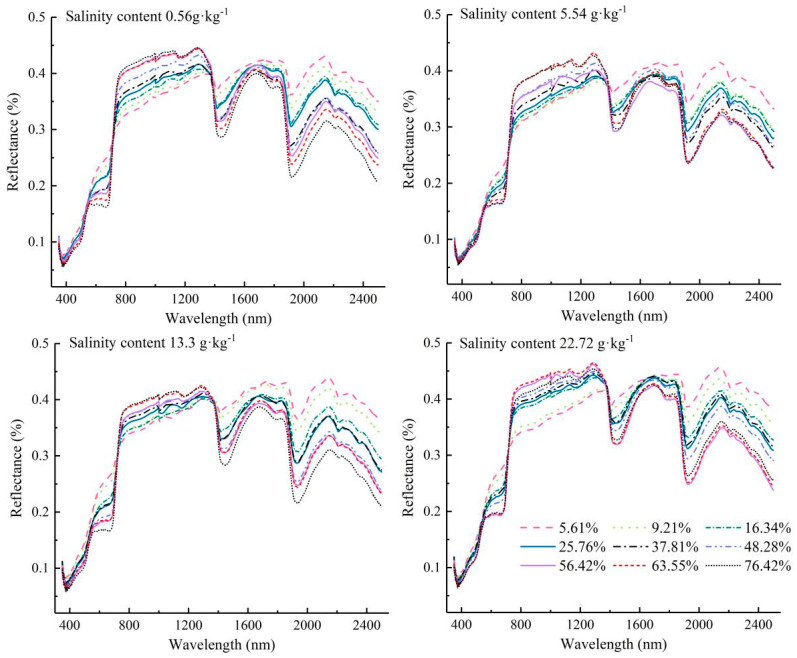
Average reflectance of salty soil with different FVCs.

**Figure 6 ijerph-20-02853-f006:**
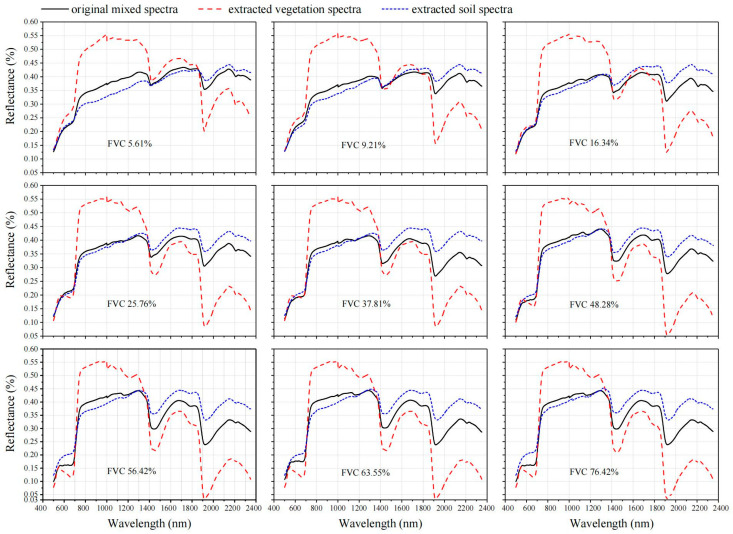
NMF-extracted spectra with different FVCs.

**Figure 7 ijerph-20-02853-f007:**
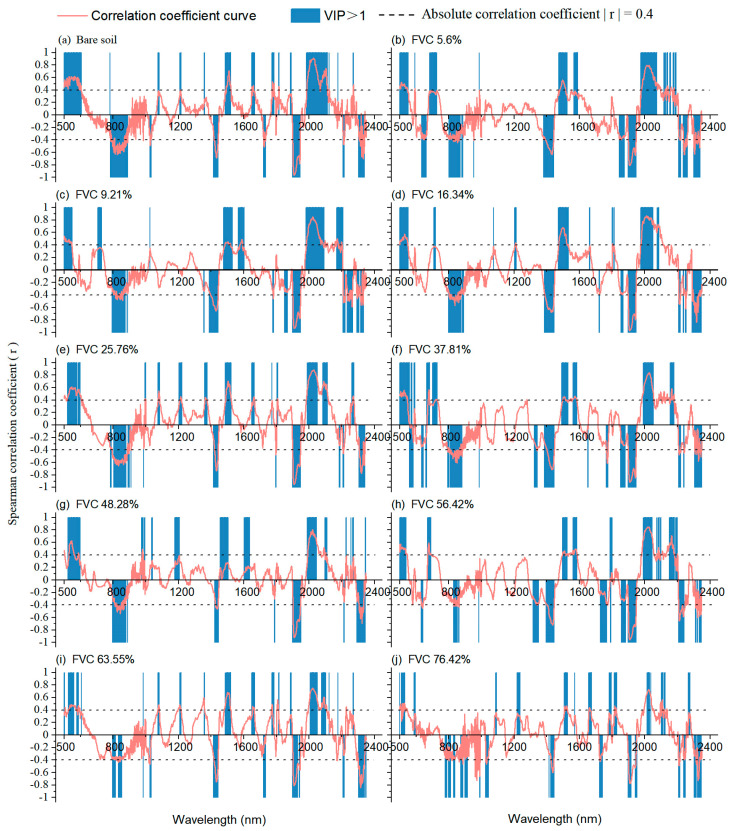
VIP scores in PLSR modeling (VIP > 1 is marked in gray) and Spearman correlation coefficient (*p* < 0.01) between SSC and mixed spectra.

**Figure 8 ijerph-20-02853-f008:**
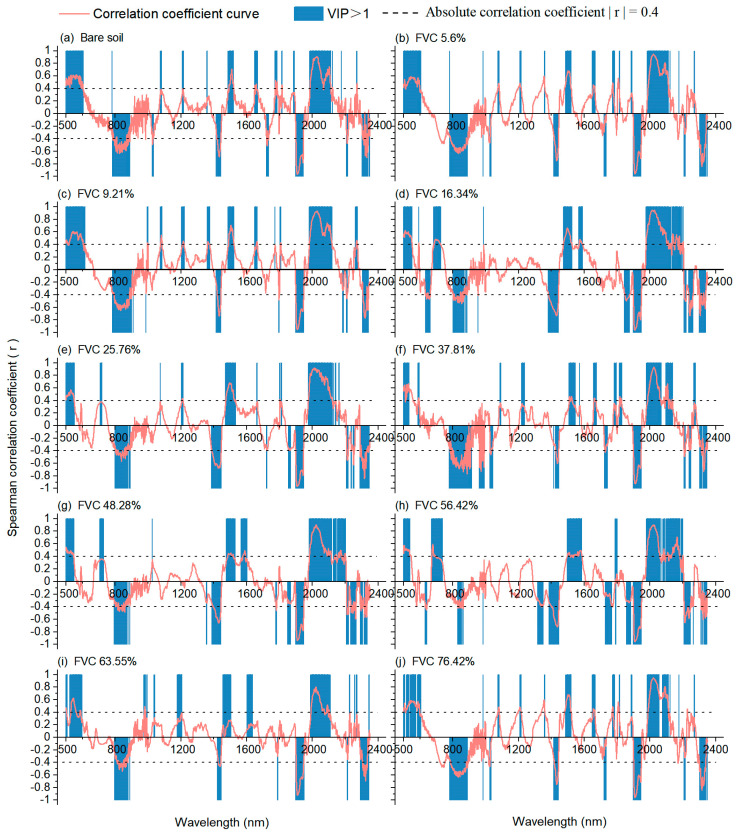
VIP scores in PLSR modeling (VIP > 1 is marked in gray) and Spearman correlation coefficient (*p* < 0.01) between SSC and NMF-extracted soil spectra.

**Table 1 ijerph-20-02853-t001:** Estimation accuracy of soil salt content by original mixed spectra at different FVCs.

FVC (%)	Calibration Set		Validation Set		
	R^2^_c_	RMSE_c_	R^2^_cv_	RMSE_cv_	RPD
Bare	0.97	2.75	0.95	2.55	2.93
5.61	0.88	3.98	0.82	3.72	2.02
9.21	0.82	4.91	0.77	4.06	1.84
16.34	0.78	5.46	0.72	4.88	1.53
25.76	0.76	5.83	0.68	5.18	1.43
37.81	0.71	6.01	0.65	5.67	1.32
48.28	0.66	5.52	0.62	5.32	1.41
56.42	0.61	6.11	0.59	5.94	1.26
63.55	0.57	6.51	0.54	6.35	1.18
76.42	0.36	8.29	0.34	7.93	0.94

**Table 2 ijerph-20-02853-t002:** Estimation accuracy of soil salt content by NMF-extracted spectra at different FVCs.

FVC (%)	Calibration Set		Validation Set		
	R^2^_c_	RMSE_c_	R^2^_cv_	RMSE_cv_	RPD
5.61	0.93	3.57	0.88	3.89	1.93
9.21	0.92	3.88	0.88	3.79	1.98
16.34	0.91	3.74	0.85	3.86	1.94
25.76	0.92	3.41	0.79	3.27	2.30
37.81	0.90	3.42	0.83	3.98	1.89
48.28	0.88	3.73	0.86	3.68	2.03
56.42	0.83	3.92	0.74	3.65	2.05
63.55	0.77	4.67	0.69	4.15	1.80
76.42	0.37	5.67	0.36	5.45	1.37

**Table 3 ijerph-20-02853-t003:** Estimation accuracy of soil salt content with multiple FVCs.

	Calibration Set		Validation Set		
	R^2^_c_	RMSE_c_	R^2^_cv_	RMSE_cv_	RPD
Mixed spectra	0.74	7.54	0.69	7.24	1.68
NMF-extracted soil spectra	0.90	4.20	0.84	5.99	2.04

## Data Availability

The data are not publicly available due to privacy.

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
