# Peer review of "Estimating Soil Salinity with Different Levels of Vegetation Cover by Using Hyperspectral and Non-Negative Matrix Factorization Algorithm"

_ijerph, 2023, doi:10.3390/ijerph20042853_

Round 1
Reviewer 1 Report
The paper deals with the estimation of the soil salinity using a combination of Hyper-spectra technique and a non-negative matrix factorization algorithm. The manuscript is written in a very good English and structure. The authors do meet the objective of the study and the results are well discussed with appropriate statistical analysis. Nevertheless, some clarifications and minor issues are needed to be done before its publication.
1/- Please add some figures in the abstract in L19-L23. This section of the manuscript dedicated to the model performance investigation is important.
2/- L121-126: Why the authors used dry soil sample and fresh wheat leaves? Is it more accurate to measure the soil water content and to consider it in the estimation process; or to use the portable spectroradiometer on site?
3/- L125-126: Why the authors used the combination 140-68 soil sample for calibration-validation?
4/- L154-171: Please indicate which software was used or which programming language was developed?
5/- L201-206: The flowchart can be moved forward the description of each step.
Reviewer 2 Report
This article focuses on the hyperspectral technique for soil salinity estimation under vegetation cover,they proposed a strategy for the model performance investigation which combines spearman correlation analysis and model variable importance projection analysis.
I think their study is meaningful, after all, I have deep experience of the laboriousness of beating soil augers between fields. The introduction section has summarized the related studies, but it is still not comprehensive and a more comprehensive summary is needed. In addition to that, the innovation of this study should be emphasized compared with related studies, and the following are some of the issues I raised:
1, Chapter 2.2 Simulated vegetation cover is not clearly explained. If the simulated cover is like Figure 2, does it restore the original local vegetation characteristics? Because the real vegetation could not look like this.
2, Rows 131-136 are the same as what Table 1 describes, so is there still a need to build the table?
3, Can Figure 3 be drawn in the same color format?
4, From Figure 5, Salinity content increased from 0.56 to 22.72, but the overall pattern in the figure is consistent, if not for taking the average value, whether there will be some interesting findings, of course, this is my little idea.
5, I have planted wheat in the Henan region, which also belongs to the Yellow River basin, and the coverage of wheat planted there is more than 76.42%, which is a problem that needs to be considered, and it is recommended to go to the actual measurement.
6, Some parts of Figure 6 have the problem of not being able to visually reflect the meaning of the text expressions, making it difficult to read.
7, It is suggested to change the legend color for Figures 7 and 8.
